# Deletion of Slc1a4 Suppresses Single Mauthner Cell Axon Regeneration In Vivo through Growth-Associated Protein 43

**DOI:** 10.3390/ijms252010950

**Published:** 2024-10-11

**Authors:** Keqiang Li, Dinggang Fan, Junhui Zhou, Ziang Zhao, Along Han, Zheng Song, Xiahui Tang, Bing Hu

**Affiliations:** 1Center for Advanced Interdisciplinary Science and Biomedicine of IHM, Division of Life Sciences and Medicine, University of Science and Technology of China, Hefei 230026, China; likeqiang@mail.ustc.edu.cn (K.L.); fdg@mail.ustc.edu.cn (D.F.);; 2Division of Life Sciences and Medicine, University of Science and Technology of China, Hefei 230001, China

**Keywords:** *slc1a4*, single Mauthner cell, Gap43, CNS axon regeneration, zebrafish

## Abstract

Spinal cord injury (SCI) is a debilitating central nervous system (CNS) disorder that leads to significant motor and sensory impairments. Given the limited regenerative capacity of adult mammalian neurons, this study presents an innovative strategy to enhance axonal regeneration and functional recovery by identifying a novel factor that markedly promotes axonal regeneration. Employing a zebrafish model with targeted single axon injury in Mauthner cells (M-cells) and utilizing the Tg (Tol056: EGFP) transgenic line for in vivo monitoring, we investigate the intrinsic mechanisms underlying axonal regeneration. This research specifically examines the role of amino acid transport, emphasizing the role of the solute carrier 1A4 amino acid transporter in axonal regeneration. Our findings demonstrate that Slc1a4 overexpression significantly enhances axonal regeneration in M-cells, whereas Slc1a4 deficiency impedes this process, which is concomitant with the downregulation of the P53/Gap43 signaling pathway. By elucidating the fundamental role of Slc1a4 in axonal regeneration and uncovering its underlying mechanisms, this study thus provides novel insights into therapeutic strategies for SCI.

## 1. Introduction

Spinal cord injury (SCI) is a highly debilitating central nervous system (CNS) disorder, with its incidence rate increasing sharply each year, leading to severe motor and sensory dysfunctions that drastically affect patients’ quality of life [1,2]. The primary challenge in spinal cord injury (SCI) repair lies in the inherent difficulty of axonal regeneration within the adult human CNS, as opposed to species that are capable of axonal regeneration within the adult CNS [3,4]. Axons in the adult mammalian CNS face significant obstacles in regenerating and re-innervating target areas due to both intrinsic and extrinsic factors [5,6]. Extrinsic factors include the presence of inhibitory molecules in the glial scar, such as chondroitin sulfate proteoglycans, which impede axonal growth [7]. Additionally, the lack of growth promoting signals in the post-injury environment further complicates regeneration efforts [8]. Intrinsic factors relate to the inability of mature neurons to re-enter a growth state and regenerate after development, due to specific physiological and molecular changes [9].

A substantial body of literature has demonstrated that amino acids play a vital role in the process of axon regeneration, essential for the recovery of nervous system function following injury [5,10,11,12]. For instance, local translation of amino acids in axons is crucial for protein synthesis necessary for axon growth and repair [10]. Specifically, neutral amino acids have been identified as particularly influential. It has been demonstrated that serine, one of the neutral amino acids, can promote axon growth and nerve regeneration through its role in the PI3K/Akt/mTOR pathway [11]. Amino acids are crucial for neuron growth, metabolism, electrical activity, and nutritional support. Slc1a4 is an amino acid transporter that facilitates the uptake of neutral amino acids, but its role in axon regeneration remains unknown [12]. Zebrafish, as a classical model animal, is highly genetically homologous to humans [13,14,15]. In recent years, due to its optical transparency, zebrafish has become an emerging model for studying axon regeneration, allowing for in vivo imaging in situ [16,17,18]. Mauthner cells (M-cells) are a pair of central neurons located in the hindbrain of zebrafish, with large axon diameters that run throughout the spinal cord [19,20,21].

M-cells exhibit strong regenerative capacity after axonal injury, allowing for the real-time visualization of axon regeneration at the single-cell level [22]. In this study, we will take full advantage of in vivo imaging of zebrafish to find out the roles of slc1a4 in the axonal regeneration. Our research aims to investigate the regulation of intrinsic factors involved in the axonal regeneration process of the CNS. Based on a new analysis, supported by data, we found that Slc1a4, among the seven members of the solute carrier family 1, exhibits the highest expression level and the most significant changes, with an increase of several folds compared to the uninjured group (Appendix A). Based on this, we speculate that Slc1a4 is crucial for axon regeneration. In vivo imaging in *slc1a4* mutant larval zebrafish constructed by CRISPR-Cas9 technology revealed that the mutation of *slc1a4* suppressed the M-cells’ axonal regeneration. Using single-cell electroporation technology, we found that the overexpression of Slc1a4 significantly promotes axonal regeneration in M-cells. RNA sequencing (RNA-seq) results indicated that the absence of Slc1a4 may inhibit regeneration through the downregulation of the P53/Gap43 axonal regeneration pathway. Real-time quantitative PCR results demonstrated that the mutation of *slc1a4* led to the downregulation of P53/Gap43 pathway relative genes. Utilizing Western blot experiments, we found that *slc1a4* mutation significantly reduced the Gap43 protein expression level. Ultimately, our study indicates that Slc1a4 plays an important role in the process of axon regeneration and provides new insights into the treatment of SCI.

## 2. Results

### 2.1. Slc1a4 Overexpression in Mauthner Cells Enhances Axon Regeneration In Vivo

To investigate the role of Slc1a4 in the regeneration of M-cells within the zebrafish CNS (Figure 1a) and to determine whether Slc1a4 functions as a critical transporter protein during the axonal regeneration of M-cells, we developed an overexpression plasmid system [23]. This setup includes CMV-GAL4-VP16/UAS-mCherry (which served as the control group) and CMV-GAL4-VP16/UAS-mCherry/UAS-*slc1a4*. First, the effectiveness of the plasmid and its expression in embryos needed to be confirmed. We utilized microinjection techniques to introduce the plasmids into embryos [13]. Screening and a quantitative polymerase chain reaction (qPCR) analysis were performed at 2 days post-fertilization (dpf). The findings revealed a significant increase in the expression level of Slc1a4 within the overexpression group. Subsequently, we aimed to further validate the impact of Slc1a4 overexpression on axonal regeneration. We mainly employed the Tg (Tol-056) line (Figure 1b), which labels M-cells, to explore the function of Slc1a4 during the axonal regeneration of these neurons [23]. With the single-cell electroporation platform developed by our lab, we transferred the overexpression plasmid system into the cell bodies of unilateral M-cells in zebrafish larvae at 4 dpf. Two days after electroporation, we carried out two-photon laser ablation right above the lesioned side of the cloacal pore. Axonal regeneration was then observed at 8 dpf using confocal imaging. A statistical analysis of the regenerative lengths indicated that M-cells, after single-cell electroporation with the Slc1a4 overexpression plasmid (Figure 1c–e), showed significantly greater regeneration lengths at 2 dpf compared to the control group. Next, we interfered with the expression of Slc1a4 to explore its impact on axonal regeneration (Figure 1f). We then proceeded to explore the effect of disrupting Slc1a4 expression on the post-injury axonal regeneration of M-cells. The larvae at 4 dpf were subjected to the inhibitor (OH-Pro), the competitive inhibitor of the Slc1a4 protein, followed by two-photon laser ablation at 6 dpf, and the regeneration length was measured at 8 dpf. The statistical analysis revealed that following the suppression of slc1a4 function, the length of axon regeneration was significantly reduced compared to the control group (Figure 1g,h). The inhibitor effect reached a threshold when the inhibitor concentration was 1 mM, and when it was greater than this concentration, the inhibition effect was comparable. This further implies that the inhibition of Slc1a4 protein function can suppress the regenerative capacity of zebrafish M-cells after injury, playing an essential role in the axonal regeneration process of these neurons.

### 2.2. Identification of slc1a4 Mutant Zebrafish

To elucidate the role of Slc1a4 in the development of M-cells and their axonal regeneration, we employed clustered regularly interspaced short palindromic repeat CRISPR-Cas9 technology to generate an *slc1a4* gene knockout zebrafish line [24,25]. This genetically modified line was created to facilitate further experimental investigations into the functional implications of Slc1a4 deficiency.

The gene knockout target was strategically located on the first exon of the *slc1a4* gene (Figure 2a,b). Through successive generations of breeding and meticulous tail-clip sequencing for identification, we successfully established a homozygous *slc1a4* knockout line. Sequencing results revealed a deletion of ten bases in the first exon, which induced a frameshift mutation. This mutation resulted in the premature appearance of a stop codon, consequently leading to the early termination of translation (Figure 2c,d).

To confirm whether the translation of the *slc1a4* gene was indeed disrupted at the protein level in this knockout line, we conducted Western blotting (WB) analyses [26]. The WB results unequivocally indicated that the expression of the Slc1a4 protein was significantly impaired in the zebrafish knockout line. This impairment was evident from the absence or drastic reduction in the protein bands corresponding to Slc1a4 in the knockout samples compared to the wild-type controls (Figure 2e,f).

These findings substantiate the successful knockout of the *slc1a4* gene and underscore the resultant deficiency in Slc1a4 protein expression. Consequently, this knockout model provides a robust platform for investigating the physiological and molecular functions of Slc1a4 in M-cell development and axonal regeneration. Future experiments will leverage this model to elucidate the mechanistic pathways through which Slc1a4 influences these critical neurobiological processes.

### 2.3. Slc1a4 Deficiency Does Not Impact Mauthner Cell Development or Motor Function

To facilitate further investigation into the impact of Slc1a4 on axonal regeneration in the *slc1a4* knockout line, we crossed this homozygous knockout line with the Tg (Tol-056) line for two consecutive generations. This breeding strategy resulted in a homozygous zebrafish line with *slc1a4* knockout (Figure 3a), marked by enhanced green fluorescent protein (EGFP) [2].

Given that growth and developmental signals can influence the length of axonal regeneration, we aimed to determine whether the absence of Slc1a4 affects the development of M-cells in juvenile fish. We first measured the entire body length of the zebrafish. There was no significant difference in body length between the control and mutant strains, suggesting that the mutation in *slc1a4* did not result in an overall length difference (Figure 3b,e). We used confocal microscopy to observe the morphology of the M-cell bodies at 6 dpf, using the cell body area as an assessment criterion (Figure 3c,f). The imaging results showed that in the absence of Slc1a4, the size of M-cell bodies at 6 dpf did not change significantly, and morphologically, there was no obvious difference between the *slc1a4* mutant and the wild-type cytosol [19]. Therefore, the deficiency of Slc1a4 did not affect the development of the M-cell bodies. Next, to rule out the impact of Slc1a4 deficiency on axonal development, we measured the length of the M-cell axons uniformly. We chose 6 dpf as the time point to rule out the possibility that the absence of Slc1a4 causes differences in the rate of axonal growth. Using the axon directly above the cloaca as the starting point (which is also the starting point of injury), we measured its extension to the tail end. This length was used for a relative comparison of the total length to determine if there were any differences between the *slc1a4* knockout group and the WT (Figure 3d,g). Imaging results showed that the axon length of M-cells was not affected by the absence of Slc1a4, being similar to that of WT, with no significant differences. In addition, juvenile fish treated with an Slc1a4 inhibitor did not exhibit significant changes in spontaneous movements compared to controls (Figure 3h,i). In addition, we treated zebrafish with an inhibitor of *slc1a4* and then examined spontaneous movements after treatment. After the assay, the results showed that the inhibitor treatment did not affect the spontaneous movements of zebrafish (Appendix A). Based on these findings, we conclude that the lack of Slc1a4 does not impact the growth and development of M-axons on the whole.

### 2.4. Slc1a4 Deficiency Suppresses Mauthner Cell Axon Regeneration and Associated Functions In Vivo

To explore the effect of Slc1a4 on regeneration after axonal damage, we followed the same timeline as before for processing the *slc1a4* mutant line (6 dpf for laser damage, 8 dpf for imaging statistical length) [27]. After the data analysis, it became evident that the regeneration length of axons significantly decreased in the absence of Slc1a4 compared to WT (Figure 4a–c). In addition, we looked at the regeneration of the *slc1a4*^+/−^ mutant lines, and after counting, it is clear that even single-stranded mutations result in differences in regeneration length (Appendix A). Previous studies have indicated that M-cells are particularly important for rapid escape responses. The C-Start is a very fast startle or escape reflex where M-cells play a crucial role [28,29,30]. Under predator pressure, zebrafish exhibit C-Start escape behavior. After receiving a stimulus, M-cells generate a strong electrical signal, which causes a strong and rapid contraction upon reaching the muscles, leading to the fish making a C-shape curve movement. Based on these principles and other studies, we constructed a C-Start escape response device for testing the function of Mauthner neurons (Figure 4d). The main indicators for assessing the recovery of Mauthner neuron function are the maximum deflection angle reached by the zebrafish after stimulation and the time taken to reach this maximum deflection angle (Figure 4e).

We performed laser ablation at 6 dpf and conducted behavioral tests at 8 dpf. After the statistical analysis, it was found that in the group where Mauthner neurons were not damaged, there were no significant differences in the time taken to reach the maximum deflection angle or the maximum deflection angle itself after stimulation (Figure 4f). However, in the knockout strains, significant changes were observed in both the time taken to reach the maximum deflection angle and the maximum deflection angle itself after the damage. This further indicates that the knockout of *slc1a4* severely inhibits the axonal regeneration ability and functional recovery of Mauthner neurons (Figure 4g–j).

In summary, the knockout of *slc1a4* did not affect the development of the M-cell bodies and axons, allowing us to exclude development as a factor in the influence of Slc1a4 on axonal regeneration. Moreover, our behavioral results also indicate that Slc1a4 does not affect the activity level of the larvae, thus ruling out spontaneous activity as an influencing factor on axonal regeneration.

### 2.5. RNA-Seq Revealed That Slc1a4 May Influence Mauthner Axons’ Regeneration through P53 Signaling Pathway

To uncover the specific molecular mechanisms affecting regeneration, we performed whole-genome transcriptome sequencing (Figure 5a) on *slc1a4*^−/−^ zebrafish and wild-type (WT) zebrafish at 4 dpf [31]. The bioinformatics analysis of the sequencing data revealed 4372 out of 24,064 core genes with significant expression changes, including 1863 upregulated and 2509 downregulated genes (Figure 5b,c).

The Gene Ontology (GO) enrichment analysis (Figure 5d) and Kyoto Encyclopedia of Genes and Genomes (KEGG) functional enrichment analysis (Figure 5e) revealed that a subset of these differentially expressed genes are primarily enriched in the P53 signaling pathway, amino acid metabolism, and regeneration processes. Research indicates that the P53 signaling pathway is associated with neuronal regeneration [32,33]. Existing research indicates that *tp53* influences the expression levels of known neuronal regeneration-related factors such as *gap43*, *rab13*, and *coronin1a*.

These findings suggest that the P53 signaling pathway may play a crucial role in mediating the effects of Slc1a4 on axonal regeneration. The differential expression of genes involved in this pathway highlights a potential molecular mechanism through which Slc1a4 deficiency impacts the regenerative capacity of Mauthner neuron axons. Further investigation into these specific genes and their interactions within the P53 signaling pathway could provide deeper insights into the molecular underpinnings of axonal regeneration and the role of Slc1a4 in this process.

### 2.6. Slc1a4 Deletion May Inhibit Mauthner Axons’ Regeneration via Gap43 Suppression

The sequencing results showed that genes such as *tp53*, *gap43*, and *rab13* all experienced a significant decrease (Figure 6a). Based on the sequencing results, it is speculated that the lack of Slc1a4 may lead to a deficiency of nutritional factors, resulting in the generation of nutritional stress signals, which in turn causes a decrease in the expression level of *tp53.* This leads to a synchronous decrease in the expression levels of genes such as *gap43* and *rab13*, thereby affecting the ability of axonal regeneration. To prove this hypothesis, we tested the expression levels of these genes. After the qPCR assay, it can be found that the expression levels of *tp53*, *gap43*, and *rab13* in the *slc1a4*^−/−^ mutant lines all underwent a significant reduction relative to the wild type (Figure 6b). After the WB assay, it can be seen that a significant reduction in the protein expression level of *gap43* occurred in the *slc1a4*^−/−^ mutant line (Figure 6c,d). It is tentatively possible to conclude that mutant strains of *slc1a4*^−/−^ may affect axon regeneration through a pathway that affects *tp53* and *gap43*.

## 3. Discussion

In this study, we investigated the role of the *slc1a4* gene in axonal regeneration using a zebrafish model with single axon injury on Mauthner cells (M-cells). Our results demonstrated that the overexpression of Slc1a4 significantly promotes axonal regeneration, while a deficiency of Slc1a4 severely impedes this process. This impairment in regeneration was associated with the downregulation of key genes in the P53 signaling pathway, including *gap43* and *rab13*. These findings suggest that Slc1a4 is a critical factor in the axonal regeneration of M-cells, potentially through its regulation of amino acid homeostasis and the P53 signaling pathway.

Both extrinsic factors and intrinsic factors are crucial for successful axonal regeneration [6,34,35,36]. Traditional models of axonal regeneration have predominantly utilized animal models such as rodents and invertebrates. These models have been instrumental in understanding the molecular and cellular mechanisms underlying axonal regeneration [34,37]. For instance, rodent models, particularly mice and rats, have been extensively employed to study spinal cord injuries and peripheral nerve regeneration. However, one of the significant challenges in studying axonal regeneration is the heterogeneity among neurons [38,39,40,41]. To address the issue of neuronal heterogeneity, our research focuses on the use of single M-cells. Dissimilar to what has been shown in a mouse model of axonal regeneration, after an M-cell was damaged, it showed strong regenerative ability. After two-photon laser damage to the axon, the regeneration length can reach about 500 μm at 2 days post axotomy, which is very convenient for the study of axons. In M-cells’ model, single-cell overexpression technology was utilized to overexpress *slc1a4*, leading to the conclusion that slc1a4 overexpression may promote axon regeneration in single M-cells.

Many studies have shown that SLC1A4, also known as ASCT1, is a neutral amino acid transporter that has been implicated in various physiological processes, including amino acid homeostasis, neurotransmission, and cellular metabolism [12,42,43,44]. Previous studies have primarily focused on its role in the CNS and its involvement in neurological disorders. For instance, mutations in Slc1a4 have been linked to developmental delay, microcephaly, and hypomyelination, highlighting its critical role in brain development and function [42]. SLC1A4 is also involved in the regulation of d-serine levels in the brain, which is essential for NMDA receptor function and neurodevelopment [12]. Despite these significant findings, the role of SLC1A4 in axonal regeneration remains largely unexplored. Our research aims to fill this gap by investigating the function of Slc1a4 in neuronal axonal regeneration. Utilizing in vivo imaging techniques, we have demonstrated that Slc1a4-mediated neutral amino acid transport may influence axonal regeneration through the P53/Gap43 signaling pathway. After treatment with an inhibitor of *slc1a4*, it was found that the regeneration of M-cells in zebrafish would be significantly inhibited. The mutation of *slc1a4* inhibits axonal regeneration and motor function recovery.

Previous studies have shown that GAP43 (Growth-Associated Protein 43) is a well-established marker of axonal growth and regeneration [45,46,47,48]. This phosphoprotein plays a crucial role in axonal outgrowth, synaptic plasticity, and neural development. Predominantly expressed in growth cones of developing neurons and regenerating axons, GAP43 is a key player in neuronal repair processes [49]. The importance of GAP43 in axonal regeneration has been highlighted through various studies. For instance, the upregulation of GAP43 is associated with axonal growth in the rat spinal cord following compression injury [50,51,52,53,54]. Building on the established role of GAP43 in axonal regeneration, our research introduces a novel finding: the knockout of *slc1a4* significantly impacts the expression of Gap43, thereby inhibiting axonal regeneration. This discovery provides new insights into the molecular mechanisms underlying axonal regeneration. It suggests that Slc1a4 is not only involved in amino acid transport but also plays a pivotal role in the regulation of key regenerative proteins like Gap43. This finding opens up potential therapeutic avenues for enhancing axonal regeneration by targeting the Slc1a4/Gap43 pathway. In conclusion, while previous studies have firmly established the role of Gap43 in axonal regeneration, this research highlights the regulatory effect of Slc1a4 and Gap43. While transcriptomic sequencing and WB have identified Slc1a4 as a potential regulator of Gap43 expression, thereby influencing axonal regeneration, the direct experimental validation of the interaction between Slc1a4 and Gap43 remains to be established [43,44,45,46].

The use of the Mauthner-cell model has indeed provided significant advantages by addressing the issue of neuronal heterogeneity. This model allows for a more controlled and precise investigation of axonal regeneration mechanisms. However, the Mauthner-cell model is inherently limited due to its specificity and uniqueness. The findings derived from this model may not be universally applicable across different neuronal types and organisms. To gain a comprehensive understanding of the role of SLC1A4 in axonal regeneration, it is essential to extend the investigation to other models. Exploring the function of Slc1a4 in diverse neuronal systems and organisms will help to validate the findings from the Mauthner-cell model and provide broader insights into its role in axonal regeneration. Such studies would involve using various in vivo and in vitro models, including rodent models, zebrafish, and other well-established systems for studying neuronal repair. This approach will help to establish the generalizability of the Slc1a4/Gap43 interaction and its implications for axonal regeneration across different biological contexts. In conclusion, while the Mauthner-cell model has provided valuable insights into the role of Slc1a4 in axonal regeneration, further research using a variety of models is necessary to fully elucidate the molecular mechanisms involved. The direct experimental validation of the interaction between Slc1a4 and Gap43, along with studies in different neuronal systems, will enhance our understanding of the pathways regulating axonal regeneration and potentially lead to the development of targeted therapeutic strategies for neuronal repair.

Despite the significant differences in neural regeneration abilities between zebrafish and mammals, particularly humans, this study has revealed the crucial role of the Slc1a4/P53/Gap43 signaling pathway in axonal regeneration in zebrafish, offering a potential target for human spinal cord injury (SCI) treatment strategies. This discovery underscores the importance of identifying and utilizing factors, such as P53 and Gap43, that promote axonal regeneration in the research of human SCI treatment to improve neurological function recovery. Future research should focus on exploring the pathways through which Slc1a4 affects axonal regeneration in mammals and further investigating the conservation of its mechanism in regulating P53 and Gap43. This will enhance our understanding of the similarities and differences in this process across different species, thereby laying a foundation for designing more effective treatment methods for human SCI.

In summary, we discovered that Slc1a4-mediated neutral amino acid transport influences Gap43 expression through the P53 signaling pathway in zebrafish. In vivo imaging revealed that Slc1a4 plays a crucial role in axonal regeneration. The loss of Slc1a4 disrupts amino acid homeostasis, leading to the accumulation of transported neutral amino acids and subsequently reducing axonal regeneration levels. These findings provide new insights into the molecular mechanisms underlying axonal regeneration and suggest potential therapeutic targets for enhancing neuronal repair in spinal cord injuries. Despite some limitations, our research contributes to a better understanding of the intrinsic factors involved in axonal regeneration and offers a foundation for future studies aimed at improving outcomes for patients with CNS injuries.

## 4. Materials and Methods

### 4.1. Zebrafish Strains and Maintenance

In this study, adult zebrafish were maintained in an aquatic system at 28.5 °C with a 14/10 h light/dark cycle (14 h light and 10 h dark cycle). Embryos were collected from natural spawning after mating male and female zebrafish in a 3:2 ratio and raised at 28.5 °C in an incubator with 5 mM NaCl, 0.17 mM KCL, 0.33 mM CaCl_2_, 0.33 MgSO_4_, and 0.1% methylene blue at pH 7.0. From 2 dpf, embryos were supplemented with 0.003% N-phenylthiourea (PTU, Sigma-Aldrich, St. Louis, MO, USA) to prevent pigmentation [55,56,57]. The transgenic line used was Tg (Tol 056: EGFP), where M-cells express enhanced green fluorescent protein. The USTC (University of Science and Technology of China) Animal Resources Center and University Animal Care and Use Committee provided the guidelines that experiments followed. Protocols needed approval by the USTC ethics committee (license number: USTCACUC1103013).

### 4.2. Genome Editing

The CRISPR-mediated genome editing for generating *slc1a4*^−/−^ mutants was implemented. Cas9 mRNA was synthesized in accordance with the appropriate plasmid (108301, Addgene, Watertown, MA, USA) through the mMessage mMachine T7 Ultra Kit (AM1345 mMESSAGE mMACHINE™ T7 ULTRA, Thermo Fisher, Waltham, MA, USA). The single-guide RNA (sgRNA) that targeted the *slc1a4* sequence was fabricated by using the aforesaid plasmids with the Megashortscript T7 kit. We delicately combined the Cas9 mRNA (300 ng/μL) with the sgRNA (40 ng/μL), and thereafter, microinjected this mixture into embryos at the one-cell stage.

### 4.3. Single-Cell Electroporation

Prior to electroporation, 4 dpf larvae were anesthetized using ethyl 3-aminobenzoic methanesulfonate (MS222, Sigma-Aldrich) and embedded in 1% low-melting agarose (Sangon, Shanghai, China) within an electroporation chamber. A micropipette tip pulled by a micropipette puller (Sutter Instrument, Novato, CA, USA) was filled with plasmids and positioned near the M-cell soma. Electric stimulation was applied to the zebrafish larvae to deliver the plasmids, with a concentration of 120 ng/μL, into the unilateral M-cell.

### 4.4. Two-Photon Axotomy

Before axotomy, anesthetized zebrafish larvae at 6 dpf were fixed in 1% low-melting agarose in a chamber. A Zeiss microscope (LSM980; Carl Zeiss, Oberkochen, Germany) equipped with two photons was utilized at a wavelength of 800 nm and an intensity ranging from 15% to 35% to ablate the M-cell axons under a 25× oil immersion lens.

### 4.5. In Vivo Imaging

Larvae were sedated with MS222 and then embedded in 1% low-melting agarose in a chamber [58]. Larvae were photographed two days after axotomy using a confocal system (FV1000; Olympus, Tokyo, Japan) and a water immersion lens (40×, 0.85 numerical-aperture objective). Z-stack images were acquired at 3 μm intervals.

### 4.6. Quantitative Real-Time PCR

Total RNA was extracted from the entire larvae through the use of RNAsio (TAKARA), and approximately 1 μg of RNA was reverse-transcribed into cDNA with the assistance of HiScript II qRT SuperMix II (Vazyme, Nanjing, China). The qPCR application was carried out in a total volume of 10 μL in a contained 5 μL of ChamQ Universal SYBR Green qPCR Master Mix and 1 μL of the cDNA template on a real-time quantification system (LightCycler 96, Roche, Pleasanton, CA, USA). The mRNA expression levels were analyzed by means of the comparative Ct relative quantification method formula 2^−∆∆CT^, with the housekeeping gene β-actin mRNA serving as an invariant control to normalize the mRNA of the target genes [59]. This was repeated three times for each sample. All the primers used are detailed in Table 1.

### 4.7. Protein Extraction and Western Blotting

Wild-type and *slc1a4*^−/−^ mutant larvae at 4 dpf were collected and lysed with an RIPA (Radio Immunoprecipitation Assay Lysis) buffer supplemented with a protease inhibitor and phosphatase inhibitor (Sangon, Shanghai, China). The lysates were centrifuged and the collected supernatant was maintained on ice. In accordance with the manufacturers’ instructions of the BCA Protein Assay Kit (Beyotime, Shanghai, China) the concentration of each protein sample was determined on a microplate reader. Samples were boiled for 4 min and run on a 10% SDS-PAGE gel along with a loading buffer (5×) and then transferred to a PVDF membrane. After incubation in 5% nonfat milk and TBST for 60 min at room temperature, the membranes were washed once with TBST and incubated with antibodies against Slc1a4 (1:1000; Proteintech, Wuhan, China) or Gapdh (1:2000; HuaAn, Wenzhou, Zhejiang, China) or β-tubulin (1:2000; GeneTex, Irvine, CA, USA) at 4 °C for 12 h. Subsequently, the membranes were incubated with secondary goat anti-rabbit antibodies (1:5000; Proteintech) for 1 h at room temperature. The blots were washed with TBST three times and visualized by the enhanced chemiluminescence (ECL) system (Thermo Fisher, MA, USA). The densities of the bands were quantified by ImageJ software (National Institutes of Health, Bethesda, MD, USA) and normalized to protein Gapdh or β-tubulin.

### 4.8. Drug Treatment

The OH-Pro (Cat.No.H54409, MCE, Shanghai, China) induces Slc1a4 downregulation. At 4 dpf, the larvae were treated with drugs by different concentrations. The concentration gradient of OH-Pro is 0.5 mM, 1 mM, and 1.5 mM. The control group is 0.1% DMSO. At 6 dpf, the axons were ablated and continued to be restored in drugs. The M-cells’ axon regeneration was observed and statistically analyzed at 8 dpf.

### 4.9. Escape Behavior Assay

The device system was comprised of a high-speed camera (1000 fps, Revealer, Hefei, China), a computer, and a loudspeaker. The 8 dpf zebrafish larvae were placed in a Petri dish with EM and moved to a platform with suitable light. The computer was linked to the loudspeaker near the Petri dish, and the high-speed camera was adjusted appropriately. Before each test, the larvae were left for 5 min without any disturbance. Movement trajectories were induced by sound stimulation of sinusoidal waves (500 Hz, 20 ms), and video acquisition was controlled by specialized software. For the injured group, unilateral M-cells were ablated with two-photon axotomy at 6 dpf before the escape behavior assay.

### 4.10. Statistical Analysis

Graphs and statistical significance were analyzed using GraphPad Prism 10.2.3 software (San Diego, USA), Adobe Photoshop CC2020, and Adobe Illustrator CC 2020. Data are presented as the mean ± standard error of the mean (SEM). Experiments were analyzed using unpaired two-tailed Student’s *t*-tests. Experiments with more than two groups were analyzed using one-way analyses of variance (ANOVAs), and experiments involving two independent variables were analyzed using two-way ANOVAs. Experiments were repeated at least three times. Differences were considered significant when * *p* < 0.05, ** *p* < 0.01, *** *p* < 0.001, and **** *p* < 0.0001. The figure legends provide all other pertinent information, such as sample size and precise statistical tests used.

### 4.11. RNA-Seq

The *slc1a4*^−/−^ lines were in the experimental group and WT was in the control group. At 6 dpf, total RNA was extracted by screening for positive expression (n = 35, with each sample repeated three times). Total mRNA was enriched using Oligo (dT) beads, and then it was fragmented into short fragments with a fragmentation buffer and reversely transcribed into cDNA using the NEB Next Ultra RNA Library Prep Kit for Illumina (Cat. No. #7530, New England Biolabs, Ipswich, MA, USA). The cDNA libraries were sequenced on the Illumina sequencing platform by Sangon Biotech (Shanghai) Co., Ltd. (Shanghai, China). In order to obtain high-quality clean reads, we filtered it using fastp (version 0.18.0) [60]. Bowtie2 (version 2.2.8) was used to remove ribosome RNA (rRNA) reads by aligning to the zebrafish rRNA database. Then, HISAT2 [61] was used to make paired-end clean reads’ map to the reference zebrafish genome (Ensembl_release109) [15]. RSEM was used to calculate the expression abundance, and it was normalized to Transcripts Per Kilobase of exon model per Million mapped reads (TPMs). A bioinformatic analysis was performed using RStudio version cd7011dc. The differentially expressed genes (DEGs) were identified using the DESeq2 package version 1.42.0 [62]. The Gene Ontology (GO) and Kyoto Encyclopedia of Genes and Genomes (KEGG) enrichment analyses of DEGs were conducted using the clusterProfiler package version 1.10.1 [63,64,65]. The enrichment significance was determined by Fisher’s exact test, and the false discovery rate (FDR) was corrected. The heatmap of DEGs was generated using the pheatmap package version 1.0.12.

## Figures and Tables

**Figure 1 ijms-25-10950-f001:**
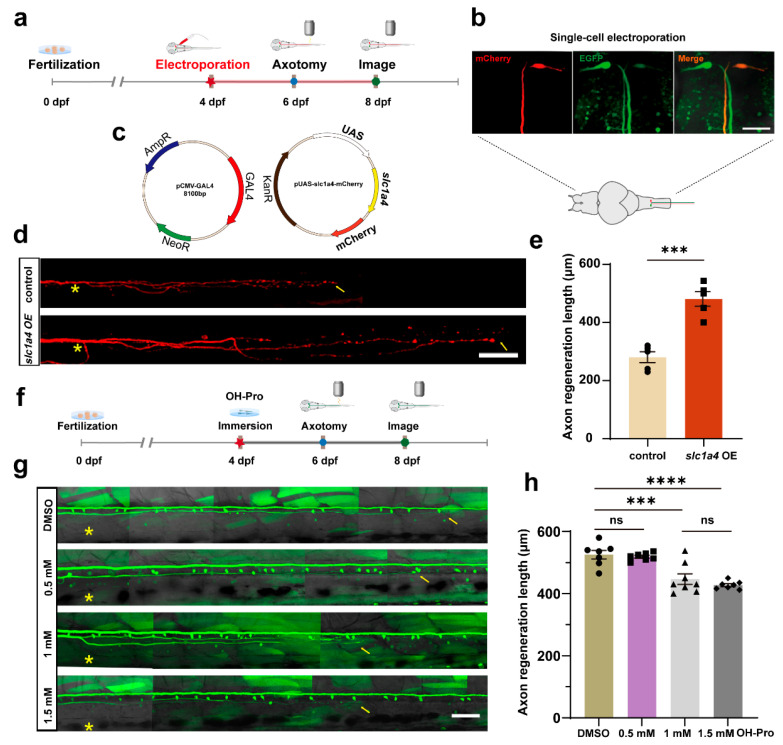
Overexpression of slc1a4 in Mauthner cells promotes axon regeneration in vivo. (**a**) Timeline of time points of electroporation, axotomy, and imaging. (**b**) Pattern diagram of electroporation and confocal images of positive expression in M-cells; three photos represent position of M-cell under 40× magnification. Scale bar, 50 μm (EGFP: labeled M-cells in Tol-056 zebrafish strain; mCherry: fluorescent reporter gene in foreign plasmid). (**c**) Schematic diagram of microinjection using two-plasmid system. (**d**) Representative diagram of confocal imaging of M-cells’ axon regeneration. Asterisk, ablation site; Arrow, regeneration endpoint location. scale bar, 50 μm (control: control; *slc1a4* OE: overexpression). (**e**) Statistical quantitative diagram of axon regeneration. Data shown as mean ± sem (control: 280.0 ± 18.8 μm, *n* = 5; *slc1a4* OE: 452.8 ± 34.0 μm, *n* = 5). Assessed by unpaired, two-tailed Student’s *t*-test. *** *p* < 0.001. (**f**) Timing of inhibitor processing, laser damage, and imaging. (**g**) Representative diagram of confocal imaging of M-cells’ axon regeneration between DMSO and inhibitor (concentration gradient, OH-Pro: 0.5 mM, 1 mM, 1.5 mM). Asterisk: ablation site. Arrowhead: axon regeneration terminal; scale bar, 50 μm. (**h**) Statistical quantitative diagram of axon regeneration. Data shown as mean ± sem (DMSO: 525.6 ± 13.9 μm, *n* = 7; 0.5 mM: 519.0 ± 4.6 μm, *n* = 7; 1 mM: 446.9 ± 17.0 μm, *n* = 8; 1.5 mM: 427.3 ± 4.879 μm, *n* = 7); data were analyzed with one-way ANOVA. *** *p* < 0.001; **** *p* < 0.0001; ns, not significant.

**Figure 2 ijms-25-10950-f002:**
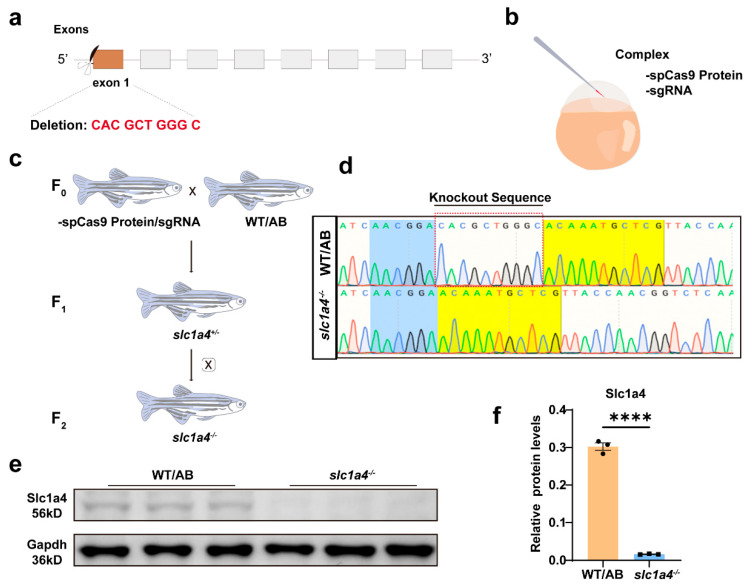
Identification of *slc1a4* mutant zebrafish. (**a**) Schematic of Cas9-sgRNA targeted site located in first exon of *slc1a4*. (**b**) Schematic of complex injected into one-cell embryos. (**c**) Procedure for obtaining *slc1a4*^−/−^ mutant lines. (**d**) Representative sequencing results of wild-type and mutated zebrafish lines. Mutant sequencing results showed that 10 bp were deleted in *slc1a4*^−/−^ zebrafish. (**e,f**) Western blotting analysis showed that Slc1a4 protein expression is inhibited in mutant group compared with wild type. Protein expressions were quantified by Image J software version 1.54f. Experiment was repeated three times with three independent samples. **** *p* < 0.0001. Assessed by unpaired *t*-test.

**Figure 3 ijms-25-10950-f003:**
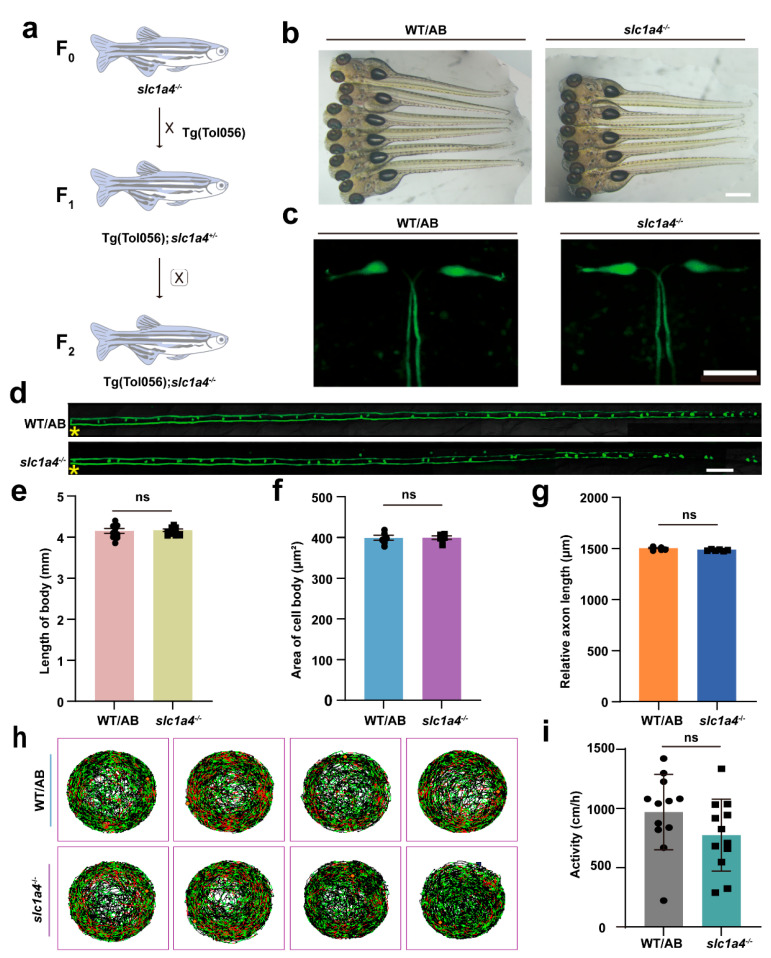
Deficiency in Slc1a4 does not affect development of M-cells and motor function of juvenile fish. (**a**) Hybridization of transgenic line: Tg (Tol 056: EGFP) and *slc1a4* mutants were crossed for two consecutive generations to obtain Tg (Tol 056: EGFP)/*slc1a4*^+/−^ and Tg (T056: EGFP); *slc1a4*^−/−^ lines. (**b**) Representative images of embryos from wild type and mutant at 6 dpf (scale bar, 500 μm). (**c**) Representative images of area of cell body (scale bar, 50 μm). (**d**) Representative images of relative axon length from cloacal pore to end (scale bar, 50 μm). Asterisk, ablation site. (**e**) Statistical quantitative diagram represents length of fish at 6 dpf. Data shown as mean ± sem (WT/AB: 4.1 ± 0.1 mm, *n* = 9; *slc1a4*^−/−^: 4.2 ± 0.0 mm, *n* = 10). Assessed by unpaired *t*-test. ns, not significant. (**f**) Statistical quantitative diagram represents area of cell body. Data shown as mean ± sem (WT/AB: 399.3 ± 5.9 μm^2^, *n* = 6; *slc1a4*^−/−^: 399.7 ± 3.9 μm^2^, *n* = 6). Assessed by unpaired *t*-test. ns, not significant. (**g**) Statistical quantitative diagram represents relative axon length. Data shown as mean ± sem (WT/AB: 1498.0 ± 6.1 μm, *n* = 6; *slc1a4*^−/−^: 1485.0 ± 3.0 μm, *n* = 6). Assessed by unpaired *t*-test. ns, not significant. (**h**,**i**) Line illustrates 6 dpf zebrafish larvae’s swimming trajectory differences from WT and *slc1a4*^−/−^ groups evaluated over 1 h. Data shown as mean ± sem (WT/AB: 1230.0 ± 70.1 cm/h, *n* = 6; *slc1a4*^−/−^: 844.2 ± 58.0 cm/h, *n* = 12). Assessed by unpaired *t*-test. ns, not significant.

**Figure 4 ijms-25-10950-f004:**
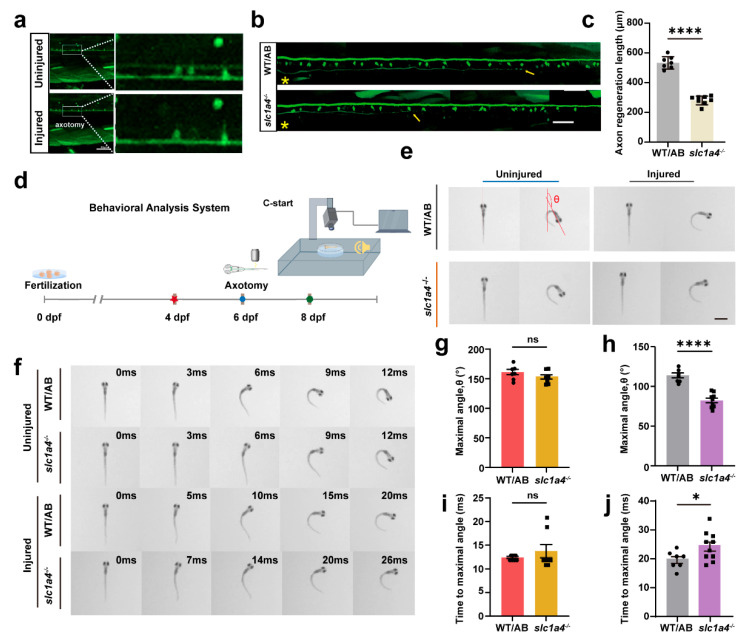
Deficiency in Slc1a4 suppresses Mauthner cell axon regeneration and relative function in vivo. (**a**) Representative images of confocal imaging of M-cell axon before and after ablation by two-photon laser above cloacal pore (scale bar, 50 μm). (**b**,**c**) Representative diagram of confocal imaging of M-cells’ axon regeneration between WT and *slc1a4*^−/−^. Data shown as mean ± sem. Scale bar, 50 μm. Assessed by unpaired *t*-test. **** *p* < 0.0001. Asterisk, ablation site; Arrow, regeneration endpoint location. (**d**) Device for testing escape behavior. (**e**) Representative images of original orientation and maximal turn angle position from WT and *slc1a4*^−/−^ zebrafish larvae in uninjured and injured groups. Red lines indicate heading direction. (**f**) Series of images of movement trajectory from WT and *slc1a4*^−/−^ zebrafish larvae in uninjured and injured groups. (**g**,**h**) Statistical diagram of maximal turn angle, θ. Data shown as mean ± sem. Uninjured: ns, not significant (WT/AB: 161.2 ± 11.89°, *n* = 7; *slc1a4*^−/−^: 152.9 ± 10.36°, *n* = 8); injured: **** *p* < 0.0001 (WT/AB: 113.9 ± 3.1°, *n* = 7; *slc1a4*^−/−^: 82.36 ± 2.917°, *n* = 10). ns, not significant. Scale bar, 1 mm. Assessed by unpaired *t*-test. (**i**,**j**) Statistical diagram of time to maximal turn angle. Data shown as mean ± sem. Uninjured, ns, not significant (WT/AB: 12.4 ± 0.2 ms, *n* = 7; *slc1a4*^−/−^: 13.75 ± 1.386 ms, *n* = 8); injured, * *p* < 0.05 (WT/AB: 19.86 ± 1.143 ms, *n* = 7; *slc1a4*^−/−^: 24.50 ± 1.586 ms, *n* = 10); ns, not significant; * *p* < 0.05. Assessed by unpaired *t*-test.

**Figure 5 ijms-25-10950-f005:**
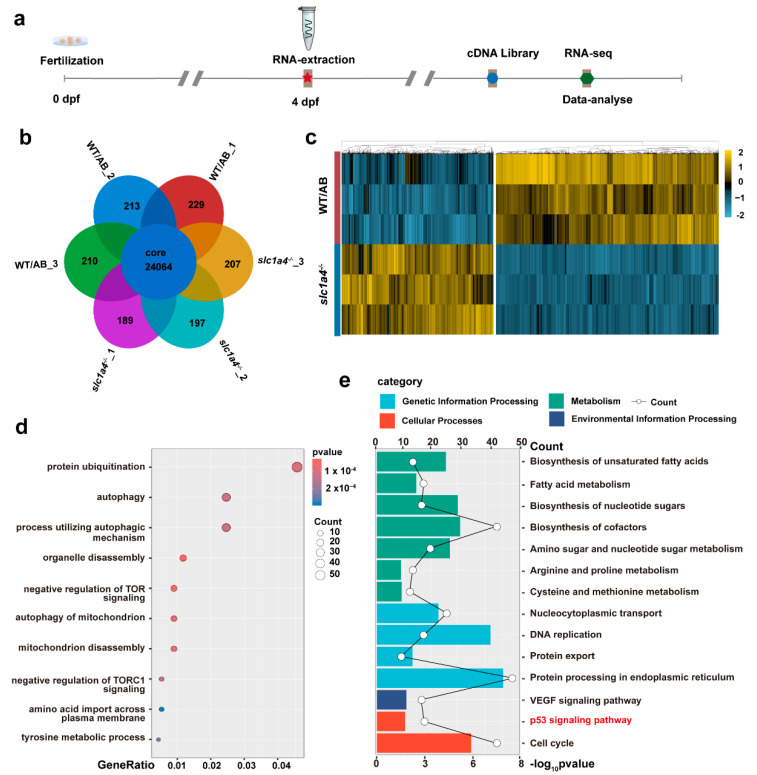
RNA-seq revealed that Slc1a4 may influence regeneration of Mauthner axons through P53 signal pathway. (**a**) Timeline of time points of RNA extraction and RNA-seq. (**b**,**c**) Heatmap shows downregulated and upregulated genes in *slc1a4* mutant zebrafish (downregulated genes: 2509; upregulated genes: 1863). (**d**) GO enrichment analysis of upregulated genes. (**e**) Enrichment for KEGG pathway analysis of downregulated genes.

**Figure 6 ijms-25-10950-f006:**
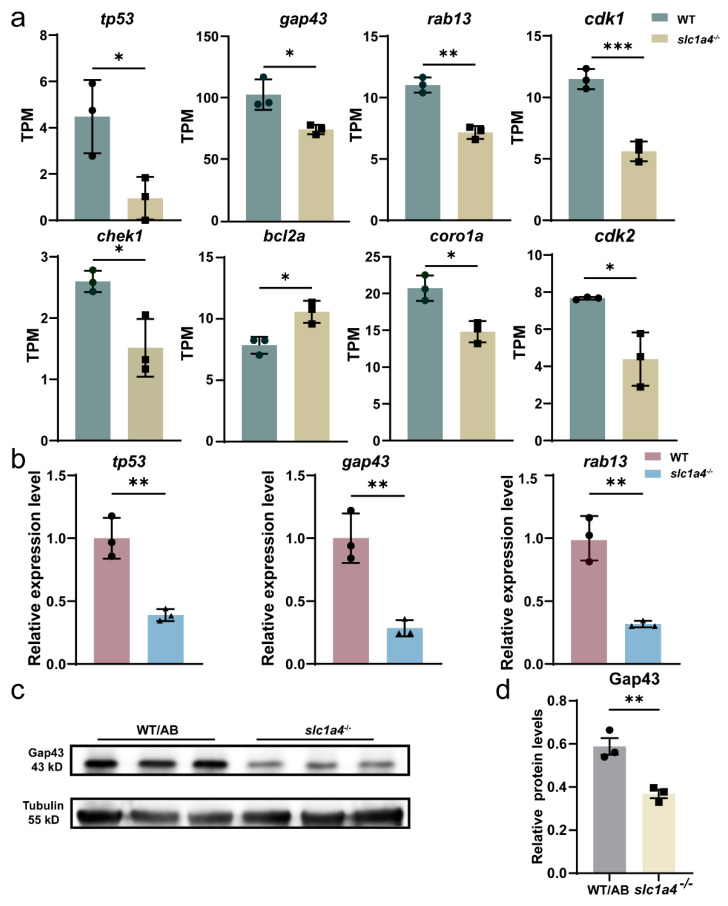
Slc1a4 deletion may inhibit Mauthner axons’ regeneration via Gap43 suppression. (**a**) RNA-seq analysis of P53 signaling pathway-related gene expression in mutant and wild-type lines. TPMs: Transcripts Per Kilobase of exon model per Million mapped reads. Assessed by unpaired *t*-test. * *p* < 0.05. ** *p* < 0.01. *** *p* < 0.001 (**b**) qPCR detection of *tp53* and *gap43* and *rab13* expression. ** *p* < 0.01. (**c**,**d**) Gap43 was decreased in knockout group as shown via Western blot quantification. Data shown as mean ± sem. Unpaired Student’s two-tailed *t*-test; ** *p* < 0.01.

**Table 1 ijms-25-10950-t001:** Primes used in qPCR experiments.

Primers	Sequences (5′-3′)
*β-actin*-qPCR-F	CATTGGCAATGAGCGTTTC
*β-actin*-qPCR-R	TACTCCTGCTTGCTGATCCAC
*tp53*-qPCR-F	TGGAGAGGAGGTCGGCAAAATCAA
*tp53*-qPCR-R	GACTGCGGGAACCTGAGCCTAAAT
*gap43*-qPCR-F	TGCTGCATCAGAAGAACTAA
*gap43*-qPCR-R	CCTCCGGTTTGATTCCATC
*rab13*-qPCT-F	GCATACTACAGAGGGGCCA
*rab13*-qPCT-R	CATTCGACTTACACCCGCTG
*slc1a4*-sgRNA-F	TAATACGACTCACTATAGGCGAAATCAACGGACACGCTGTTTTAGAGCTAGAAATAGC
*slc1a4*-sgRNA-R	AGCACCGACTCGGTGCCACT
*slc1a4*-qPCR-F	TACCATCATCCCTAGCCCGA
*slc1a4*-qPCR-R	ATGGAGGAGAAGAAGAGCGAAATCA

## Data Availability

All data generated or analyzed during this study are included in this published article and its Appendix A.

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
