# Peer review of "Deletion of Slc1a4 Suppresses Single Mauthner Cell Axon Regeneration In Vivo through Growth-Associated Protein 43"

_ijms, 2024, doi:10.3390/ijms252010950_

Round 1

Reviewer 1 Report

Comments and Suggestions for Authors

In this manuscript, the authors describe the role of the serine transporter (Slc1a4) in axonal regeneration in a zebrafish model and associate it with the P53/Gap43 signaling pathway. The methodology is quite novel for axonal growth studies, as it combines gene editing, molecular biology methods, and predator escape behavior in the zebrafish model of neuronal injury. Even though this model deviates from traditional rodent models of SCI, it seeks to answer fundamental questions about neuronal regeneration after traumatic injury.

The role of other important cellular processes, such as glial cells or immune cells, is not taken into account in this approach. It is important for the reader that the authors explain their opinion on this matter in the Discussion section.

Some minor observations are:

_ One repeated sentence: lines 48 and 50

_ Spelling correction of Fig. 1b

_ TPM meaning in Fig. 6

Author Response

Comments 1: [The role of other important cellular processes, such as glial cells or immune cells, is not taken into account in this approach. It is important for the reader that the authors explain their opinion on this matter in the Discussion section.]

Response 1: [We considered the roles of glial cells and immune cells in the regeneration of hair follicle cells, but the strength of our model lies in thoroughly investigating the correspondence between single axon regeneration phenotypes and the neuronal soma, as well as the regulatory roles of endogenous factors in neuronal axon regeneration.] Thank you for pointing this out. I/We agree with this comment.

Comments 2: [Some minor observations are:

_ One repeated sentence: lines 48 and 50

_ Spelling correction of Fig. 1b

_ TPM meaning in Fig. 6]

Response 2: Thank you for pointing this out. We agree with this comment. Therefore, I have removed the repetitive paragraphs, and corrected the spelling errors in Fig 1. Additionally, the meaning of TPM in Fig 6 has been clearly described in the figure caption.

Reviewer 2 Report

Comments and Suggestions for Authors

The authors utilized zebrafish model with targeted single axon injury in Mauthner cells (M-cells) to study axonal regeneration and, in particular, role of Slc1a4 transporter in axonal regeneration in zebrafish. They demonstrated that Slc1a4 Overexpression in Mauthner Cells Enhances Axon Regeneration in vivo in zebrafish. They generated a homozygous zebrafish line with slc1a4 knockout, labeled by enhanced GFP, and demonstrated that Slc1a4 loss does not affect M-cells development and the motor function of juvenile zebrafish, but suppresses M-cells axon regeneration in vivo, perhaps via P53 signaling pathway or via influencing Gap43.

I recommend replacing «emphasizing the slc1a4 gene within the solute carrier family 1, and its critical importance in axonal regeneration” with something along the lines “emphasizing the role of Solute Carrier 1A4 amino acid transporter in axonal regeneration”.

The primary challenge in SCI repair lies in the inherent difficulty of axonal regeneration within the adult CNS” – in adult humans, as there are species capable of axonal regeneration within adult CNS.

Then talking about “intrinsic and extrinsic factors preventing mature neurons from regeneration”, the authors write “Intrinsic factors involve the poor regenerative capacity of mature neurons”. In other words, they are saying that the factor preventing neurons from regeneration is their inability to regenerate. In my opinion it’s the same as if the authors stated that neurons cannot regenerate because they cannot regenerate.

Given the differences of molecular mechanisms of neuroregeneration post-SCI in zebrafish and humans, can the authors add one more sub-chapter into Discussion, a sub-chapter discussing how transferable are all these new findings from this article to clinic and to human post-SCI therapies?

Author Response

Comments 1: [I recommend replacing «emphasizing the slc1a4 gene within the solute carrier family 1, and its critical importance in axonal regeneration” with something along the lines “emphasizing the role of Solute Carrier 1A4 amino acid transporter in axonal regeneration”.]

Response 1: [I have made the modifications according to your suggestions.]  

Comments 2: [“The primary challenge in SCI repair lies in the inherent difficulty of axonal regeneration within the adult CNS” – in adult humans, as there are species capable of axonal regeneration within adult CNS.]

Response 2: [The primary challenge in spinal cord injury (SCI) repair lies in the inherent difficulty of axonal regeneration within the adult human CNS, as opposed to species that are capable of axonal regeneration within the adult CNS [3,4]]. We have made the modifications according to your suggestions. The revised paragraph is located in lines 30-33.

Comments 3:[Then talking about “intrinsic and extrinsic factors preventing mature neurons from regeneration”, the authors write “Intrinsic factors involve the poor regenerative capacity of mature neurons”. In other words, they are saying that the factor preventing neurons from regeneration is their inability to regenerate. In my opinion it’s the same as if the authors stated that neurons cannot regenerate because they cannot regenerate.]

Response 3: [Intrinsic factors relate to the inability of mature neurons to re-enter a growth state and regenerate after development, due to specific physiological and molecular changes. [9]] Thank you for raising the question. I have made the modifications, and the revised paragraph is located in lines 38-44.

Comments 4:[Given the differences of molecular mechanisms of neuroregeneration post-SCI in zebrafish and humans, can the authors add one more sub-chapter into Discussion, a sub-chapter discussing how transferable are all these new findings from this article to clinic and to human post-SCI therapies?]

Response 4: [Despite the significant differences in neural regeneration abilities between zebrafish and mammals, particularly humans, this study has revealed the crucial role of the Slc1a4/P53/Gap43 signaling pathway in axonal regeneration in zebrafish, offering a potential target for human spinal cord injury (SCI) treatment strategies. This discovery underscores the importance of identifying and utilizing factors, such as P53 and Gap43, that promote axonal regeneration in the research of human SCI treatment to improve neurological function recovery. Future research should focus on exploring the pathways through which Slc1a4 affects axonal regeneration in mammals and further investigating the conservation of its mechanism in regulating P53 and Gap43. This will enhance our understanding of the similarities and differences in this process across different species, thereby laying a foundation for designing more effective treatment methods for human SCI.] I have added this paragraph in the discussion, located in lines 387-398.

Reviewer 3 Report

Comments and Suggestions for Authors

The use of Slc1a4 in axon regeneration represents a novel approach, and the justification for using the zebrafish model is well-founded. The combination of CRISPR technology, in vivo imaging, and qPCR techniques adds significant depth to the study's findings. Figures and images are well-labeled, making it easier for the reader to follow the experimental design. In my professional assessment, the methods appear robust, with no major flaws. However, I have several minor suggestions that could improve the manuscript and make it ready for publication:

1.Hypothesis Clarification: The hypothesis that Slc1a4 is crucial for axon regeneration is not clearly stated at the beginning of the manuscript. It would benefit the reader if a more focused hypothesis or research question was introduced early in the Introduction. Include the objectives of the study within the introduction to provide a clear direction.

2. Sample Size and Statistical Considerations: While statistical analyses are conducted, the manuscript does not explain how sample sizes were determined, nor does it address the potential for type I and type II errors. It would be helpful to include a justification for the sample size and any measures taken to control these errors.

3.Reference to Recent Studies: Although the study provides adequate references for zebrafish models, key recent studies on axonal regeneration using other models (e.g., rodents) are under-represented. Consider adding more references in the Discussion section, particularly in the following areas:

Insert references in:

"Traditional models of axonal regeneration have predominantly utilized animal models such as rodents and invertebrates. These models have been instrumental in understanding the molecular and cellular mechanisms underlying axonal regeneration." – no references

Insert references in:

"Similar to what has been shown in a mouse model of axonal regeneration, after M-cell damage, it demonstrated strong regenerative ability."

Add more references to justify using "many" rather than just two in:

"Many studies have shown that SLC1A4 (also known as ASCT1) is a neutral amino acid transporter implicated in various physiological processes, including amino acid homeostasis, neurotransmission, and cellular metabolism [12,41]."

In the statement, "Previous studies have shown that GAP43 (Growth Associated Protein 43) is…", there is only one reference for prior studies. Please include additional references to strengthen the claim.

4.Be sure to define acronyms like "USTC" (University of Science and Technology of China) and any other terms that are not immediately explained within the text.

5.Conclusions and Limitations: The conclusion and limitations sections are somewhat blended. It would be more effective to separate the conclusions paragraph from the limitations within the Discussion section. This will enhance the clarity and readability of the manuscript’s final points.

Author Response

Comments 1: [Hypothesis Clarification: The hypothesis that Slc1a4 is crucial for axon regeneration is not clearly stated at the beginning of the manuscript. It would benefit the reader if a more focused hypothesis or research question was introduced early in the Introduction. Include the objectives of the study within the introduction to provide a clear direction.]

Response 1: [Thank you very much for your comments on this study. In response to your questions, I have highlighted the crucial role of Slc1a4 in axonal regeneration in lines 62-63 and 72-73.]

Comments 2: [Sample Size and Statistical Considerations: While statistical analyses are conducted, the manuscript does not explain how sample sizes were determined, nor does it address the potential for type I and type II errors. It would be helpful to include a justification for the sample size and any measures taken to control these errors.]

Response 2: [Due to a previous writing error, the significant difference may have been incorrectly described.The sample size for this experiment was primarily determined by referencing the sample sizes used in previously published literature related to hair follicle cell regeneration.DOI:10.1016/j.expneurol.2024.114715;10.1007/s00018-024-05117-2;10.1186/s40478-022-01484-8]

Comments 3: [Reference to Recent Studies: Although the study provides adequate references for zebrafish models, key recent studies on axonal regeneration using other models (e.g., rodents) are under-represented. Consider adding more references in the Discussion section, particularly in the following areas:]

Response 3:[Thank you for pointing out the issue of insufficient literature citations. I have introduced more references in the sections where citations were lacking.] 

Comments 4: [Be sure to define acronyms like "USTC" (University of Science and Technology of China) and any other terms that are not immediately explained within the text.]

Response 4: [We have written out the abbreviations in full.]

Comments 5: [Conclusions and Limitations: The conclusion and limitations sections are somewhat blended. It would be more effective to separate the conclusions paragraph from the limitations within the Discussion section. This will enhance the clarity and readability of the manuscript’s final points.]

Response 5: [Thank you for your detailed review comments and suggestions on my manuscript. I have carefully considered your advice on separating the conclusion and limitations sections to enhance the clarity and readability of the document. During the writing process, I intentionally integrated these two sections to allow readers to grasp the main findings and limitations of the research within a coherent narrative. This approach aims to provide a more focused and comprehensive perspective, facilitating a better understanding of the study as a whole. Of course, I will further review and revise the manuscript according to your suggestions, ensuring that the presentation of the conclusions and limitations is as clear and understandable as possible to meet the needs of a broad readership.Thank you once again for your valuable feedback and support for my research work.]